# Targeted Isolation of Rubrolides from the New Zealand Marine Tunicate *Synoicum kuranui*

**DOI:** 10.3390/md18070337

**Published:** 2020-06-27

**Authors:** Joe Bracegirdle, Luke J. Stevenson, Michael J. Page, Jeremy G. Owen, Robert A. Keyzers

**Affiliations:** 1School of Chemical and Physical Sciences, Victoria University of Wellington, Wellington 6012, New Zealand; joe.bracegirdle@vuw.ac.nz; 2Centre for Biodiscovery, Victoria University of Wellington, Wellington 6012, New Zealand; luke.stevenson@vuw.ac.nz (L.J.S.); jeremy.owen@vuw.ac.nz (J.G.O.); 3Maurice Wilkins Centre for Molecular Biodiscovery, Auckland 1142, New Zealand; 4School of Biological Sciences, Victoria University of Wellington, Wellington 6012, New Zealand; 5National Institute of Water & Atmospheric Research (NIWA), P.O. Box 893, Nelson 7010, New Zealand; mike.page@niwa.co.nz

**Keywords:** rubrolide, GNPS, *Synoicum kuranui*, antibacterial

## Abstract

Global natural products social (GNPS) molecular networking is a useful tool to categorize chemical space within samples and streamline the discovery of new natural products. Here, we demonstrate its use in chemically profiling the extract of the marine tunicate *Synoicum kuranui,* comprised of many previously reported rubrolides, for new chemical entities. Within the rubrolide cluster, two masses that did not correspond to previously reported congeners were detected, and, following MS-guided fractionation, led to the isolation of new methylated rubrolides T (**3**) and (*Z*/*E*)–U (**4**). Both compounds showed strong growth inhibitory activity against the Gram-positive bacteria *Bacillus subtilis*, with minimum inhibitory concentration (MIC) values of 0.41 and 0.91 μM, respectively.

## 1. Introduction

Molecular networking (MN) through the global natural products social molecular networking (GNPS) platform [1] is a cutting-edge contemporary tool for the screening of marine natural product extracts [2]. In recent years, the tool has been used to direct the isolation of many single metabolites (particularly peptides, such as pagoamide) [3]; however, it has also proven extremely powerful in the discovery of new congeners within molecular classes or families [4,5]. MN links compounds (nodes) based on spectral alignment of their MS^2^ spectra, therefore, many metabolites within a structural class in an extract generate large constellations comprising many interconnected nodes. These clusters stand out from the noise in a large network, prioritizing samples for more detailed investigation (Appendix A). MN has also proved extremely useful for the detection of previously unreported metabolites in extensively well studied families of compounds, or from heavily investigated taxa [6,7,8,9].

As part of our continuing investigation of the secondary metabolites produced by marine invertebrates collected from the South Pacific [10,11,12,13], we recently reported the first study utilizing MN to examine the chemical constituents of various Tongan marine tunicates [14]. The major constellation of the network consisted of three linked clusters of nodes, and prioritization of the fractions containing lamellarin sulfates resulted in the characterization of the first new sulfated lamellarins reported in over 20 years [14]. Tunicates have been an excellent source of natural products to date, including the FDA-approved plitidepsin (Aplidin^®^) and trabectedin (Yondelis^®^); however, the number of new tunicate-derived compounds reported annually is decreasing, with 45% less reported in 2017 compared to 2016 [15].

Here, we explored the chemical space occupied by metabolites of an extract of the New Zealand tunicate *Synoicum kuranui*. The genus *Synoicum* is one of nine currently accepted genera of the tunicate family Polyclinidae, comprising 82 different species [16], including the source of the potent cytotoxic palmerolides, first isolated by Baker and co–workers from an Antarctic *S. adarenum* sample [17], as well as numerous alkaloids [18,19,20]. *Synoicum* also commonly harbors non-nitrogenous polyaromatic butenolides, led by the rubrolides, which are often co–isolated with other butenolide-containing families, such as prunolides and cadiolides. Like many alkaloids, these are formed by the condensation of multiple aromatic amino acid-derived metabolites [21], but do not themselves contain nitrogen. The rubrolides were first isolated from a Canadian (Queen Charlotte Islands, British Columbia) tunicate *Ritterella rubra* [22], with the full natural panel now >20 compounds after subsequent isolations from tunicates *S. globosum* [23], *S. blochmanni* [24], two unidentified *Synoicum* species [25,26], *Pseudodistoma antinboja* [27] and the marine-derived fungus *Aspergillus terreus* [28]. Given the significant number of known compounds within the class, finding new congeners is a potentially difficult exercise, however, MN can provide assistance for this even within well studied organisms.

## 2. Results and Discussion

The bright orange/red tunicate *S. kuranui* (Appendix A; Millar, 1982) [29], was collected from Great Barrier Island in the Hauraki Gulf, NZ, at depth of 21 m with SCUBA as part of the National Institute of a Water & Atmospheric Research (NIWA) collection. This tunicate is native to NZ waters, first recorded from Great Barrier Island, and has also been observed around the North Cape and Fiordland [30]. With no previously reported investigations of the species, this organism was included in a mass spectrometric screen of South Pacific tunicates, visualized using GNPS and Cytoscape (Appendix A). Constellations consisting of nodes made up from only one organism are a clear lead for similarly structured secondary metabolites. In the network, the second largest constellation consisted of nodes solely from the *S. kuranui* extract (in pink, constellation A), and thereby highlighted the tunicate for further investigation.

The methanolic extract was fractionated with HP-20^®^ (poly(styrene-divinylbenzene) copolymer (PSDVB)) resin using mixtures of H_2_O and acetone, and was analyzed by ^1^H NMR spectroscopy and mass spectrometry. The ^1^H NMR spectrum revealed numerous peaks of interest in the aromatic region, and a notable absence of peaks typically associated with fats and primary metabolites in the 0–2 ppm range (Appendix A). The sample was then further purified using reversed–phase HPLC to afford the known rubrolides A and B (**1** and **2**) and new rubrolides T and U (**3** and **4**) (Figure 1).

The major compound, rubrolide A (**1**), was isolated as a yellow film, and HRESIMS analysis gave a deprotonated molecular ion cluster at *m/z* 590.7075 indicative of the molecular formula C_17_H_7_O_4_Br_4_ (calcd. 590.7083). The ^13^C NMR spectrum consisted of 13 deshielded resonances, while the ^1^H NMR spectrum showed four deshielded singlet methine signals, with two integrating for two (*δ*_H_ 7.77 and 8.05) and one (*δ*_H_ 6.35 and 6.55) relative protons each. This data was used to search the MarinLit database [31], with rubrolide A (**1**) being the clear hit both spectroscopically and from a taxonomic perspective. A comparison of the NMR data from the original isolation confirmed this [22]. A minor compound, also isolated as a yellow film, was subject to HRESIMS analysis, which detected a deprotonated molecular ion cluster at *m/z* 624.6694 indicative of the molecular formula C_17_H_7_O_4_Br_4_Cl (calcd. 624.6694). This was dereplicated as rubrolide B (**2**), after the matching of MS and ^1^H NMR data [22].

With annotation of the structural class of the major metabolites deduced from the 75% acetone HP-20^®^ fraction, molecular networking was then used to probe the remaining two *S. kuranui* fractions for new unreported congeners, along with dereplication of other known minor metabolites. From the major constellation generated by the analysis of the remaining 30% and 100% acetone fractions, compounds **1** and **2**, and 10 other nodes were putatively annotated, based on the precursor ion matching the *m/z* of a previously reported rubrolide (Figure 2). Based on the connections between nodes and precursor ions, two previously unreported rubrolide masses were detected, with the molecular formulae C_18_H_10_O_4_Br_4_ and C_18_H_11_O_4_Br_3_. These ions were then used to prioritize the HPLC fractions for further purification.

Through a targeted isolation procedure guided by MS, a new analogue named rubrolide T (**3**), was isolated as a yellow film. Analysis of the deprotonated molecule at *m/z* 604.7254 in the HRESIMS indicated the molecular formula C_18_H_9_O_4_Br_4_ (calcd. 604.7240), consisting of 12 double bond equivalents (DBE). The presence of four bromine atoms was clear from the 1:4:6:4:1 quintet isotopic distribution pattern (Appendix A). The ^13^C NMR spectrum consisted of one methoxy and 13 deshielded resonances, including an α,β–unsaturated ester carbonyl at *δ*_C_ 168.5, which, together, indicated an element(s) of symmetry within the molecule. The ^1^H NMR spectrum showed a methoxy (*δ*_H_ 3.83) and four aromatic singlets, two integrating for two relative protons each (*δ*_H_ 7.59 and 8.15), and the other two (*δ*_H_ 6.21 and 6.44) for one proton, respectively. No COSY correlations were observed, therefore, it was likely no protons were vicinal to another.

The two-proton aromatic methine *δ*_H_ 8.15 (H-2’’/H-6’’) showed correlations to *δ*_C_ 134.2 in both the HSQC and HMBC spectra, thus, was indicative of two methines symmetrically substituted about a phenyl ring. This proton resonance also showed HMBC correlations to *δ*_C_ 153.1, 117.7 and 108.2. The correlation to an oxygenated carbon at *δ*_C_ 153.1 (C–4’’) was shared with that of methoxy *δ*_H_ 3.83, which must be on the same ring. Owing to the presence of the singlet methine and the symmetry required, these three substituents must be *meta* to one another. As the carbon *δ*_C_ 108.2 (C–6) is protonated (HSQC; *δ*_H_ 6.44), and does not show a COSY correlation to *δ*_H_ 8.15, it must be outside the aromatic ring, thus, a pair of brominated carbons were assigned to positions C–3’’/C–5’’ (*δ*_C_ 117.7), and substructure I was deduced (Figure 2). An analogous workflow starting from the other two-proton aromatic methine (*δ*_H_ 7.59) also derived a 1,3,4,5–tetrasubstituted phenyl ring, however, with no methyl substitution at the C–4’phenol, giving rise to substructure II. The remaining proton singlet (*δ*_H_ 6.21) was assigned to C–3 of the central butenolide ring, as it had the only correlation to ester carbonyl C–2 (*δ*_C_ 168.5). H–3 also shared HMBC correlations to quaternary carbon *δ*_C_ 156.6 with both H–6 and H–2’/6’, which assigned this as C–4, and, consequently, *δ*_C_ 148.8 to oxygenated C–5 (Figure 3). This molecule differs from the major compound rubrolide A **1** solely by the presence of a methoxy group at C–4’’, and as a new member of the class, is termed rubrolide T (Table 1 and Table 2).

It was noted that when two chromatographic fractions, each containing ions at *m/z* 526.8143 (HRESIMS), were purified by HPLC, there were two separate UV–detected peaks with identical ion masses observed in each sample. Upon ^1^H NMR analysis, both isolated HPLC peaks had identical NMR resonances, however, the resonances differed in relative intensities, suggesting the presence of two isomers. This suggested both of the distinct HPLC fractions contained the same two separate compounds. This has been observed before for rubrolides O, P, Q, and also cadiolide F [25,32], all of which all exist as two separable but interchangeable isomers, differing by their Δ_5,6_ geometry. All previous examples have been mono–brominated on the benzylidene ring, and are often methylated at the phenol. 

The major rubrolide U isomer (*Z*–**4**) was isolated as a yellow film, and HRESIMS analysis detected the deprotonated molecule at *m/z* 526.8143, indicative of the molecular formula C_18_H_10_O_4_Br_3_ (calcd. 526.8135). The presence of three bromine atoms was clear from the 1:3:3:1 quartet isotopic distribution pattern (Appendix A). The ^13^C NMR spectrum showed a similar pattern of peaks as **3**, including the single methoxy resonance, however *Z*–**4** has two extra proton signals and with one less bromine in the elemental formula, an element of symmetry was broken (Table 1). In the ^1^H NMR spectrum, the coupling pattern of the peaks at *δ*_H_ 7.19 (d, 8.7 Hz), 7.81 (dd, 8.7, 2.2 Hz) and 8.11 (d, 2.2 Hz) was characteristic of a 1,2,4–trisubstituted aromatic ring, while the singlet at *δ*_H_ 7.56, integrating for two relative protons, indicated a 1,3,4,5–tetrasubstituted structure (Table 2). 

The methoxy group at *δ*_H_ 3.90 showed an HMBC correlation to C–4’’ (*δ*_C_ 155.5), which also correlated with the doublet of doublets H–2’’ and the *meta*–coupled doublet H–6’’ (*δ*_H_ 7.81 and 8.11, respectively). It also showed a NOE correlation (ROESY experiment) to the *ortho*–coupled doublet H–3’’ (*δ*_H_ 7.20) only, thus the methoxy group is located on the 1,2,4–trisubstituted aromatic ring. The structure was confirmed as **4** by HMBC correlations between H–6 (*δ*_H_ 6.37) and both C–2’’ and C–6’’ (*δ*_C_ 134.0 and 131.3). The single bromination of the benzylidene unit is consistent with all other congeners isolated as mixtures of *E*/*Z* geometric isomers. This substitution pattern represents a new structure of the class and was therefore named rubrolide U. 

The remaining peaks in the ^1^H NMR spectrum were assigned to the minor rubrolide U isomer (*E*–**4**), based on relative peak intensity of the resonances in the two separate HPLC fractions, one enriched in *E*–**4** and the other *Z*–**4**, in conjunction with chemical shift values and NOE (ROESY) (Figure 4) data. For *E*–**4**, the two aromatic rings are closer in space, which results in increased shielding of the aromatic signals compared to *Z–*
**4**, where they point away from each other. (Figure 3) In both cases, H–3 showed NOE correlations to H–2’/6’, however for the *Z* isomer H–6 correlated to H–2’/6’, H–2’’ and H–6’’, whereas it only correlated to H–2’’ and H–6’’ for the *E* isomer. A correlation between H–2’/6’ and H–2’’ was also observed for the *E* isomer but absent for *Z,* consistent with the two aromatic rings in a closer proximity and placing the sterically demanding Br atom away from the molecular core.

Although the two HPLC samples were enriched for either *E*– or *Z*–**4**, over time, the proportion of the peaks corresponding to *Z*–**4** increased, suggesting isomerization to this compound. The *Z* isomer is favored, due to the reduction in steric repulsion between the two aromatic rings that are set co–planar to each other by the extended conjugation across the molecule. When the rubrolides are biosynthesized (*vide infra*), it is hypothesized that a final deprotonation forms Δ_5,6_, where both isomers could be formed if the conformation of the benzylidene ring allows it (Scheme 1). The electron density of the aromatic benzylidene group will help stabilize the positive charge formed in the conversion, particularly with the electron donating methoxy group at the *para* position. Of the four previous compounds reported to display this phenomenon, three also contained the methoxy group on the ring. In fact, the demethylated congeners of rubrolide Q and cadiolide F were not reported to be isolated as *E* isomers, therefore, it is likely a combination of both factors contributing to carbocation stability. No conversion from the *Z* to the *E* isomer has been observed.

Although there are no reports formally investigating the biosynthesis of the marine derived polyaromatic butenolides, a related butenolide isolated from the fermentation broth of *Aspergillus terreus* var. *africanus* IFO 8835, was shown, through isotope incorporation studies, to be biosynthesized from the condensation of two molecules of tyrosine, and subsequent halogenation/methylation via an α–keto acid [33,34]. These biosynthetic studies led Miao and Andersen to propose a biogenesis for the rubrolides (Scheme 2) [35]. Unlike the tunicate amino acid-derived alkaloids, the rubrolides do not contain the amino nitrogen, therefore the first step is the formation of the α–keto acid **6** via transamination of **5**. Through an Aldol–type condensation, two molecules form a reactive intermediate that then undergoes oxidative decarboxylation and reduction–dehydration to form the previously reported rubrolides G and H (**7** and **8**), after halogenation [22]. To form the remaining rubrolides, and by extension **3** and **4**, this reactive intermediate would then undergo another dehydration, halogenation and methylation. The presence of interconverting *E* and *Z* isomers with the same halogenation patterns suggest that halogenation occurs before the dehydration reaction.

All of compounds **1**–**4** showed strong inhibitory activity against *B. subtilis*, while no activity was detected against *Escherichia coli* at 128 μg mL^−1^ (Table 3). Many previously reported rubrolides have shown antimicrobial and cytotoxic activity against a range of targets, where a general trend has been observed that phenol methylation reduces activity in both synthetic [36] and a naturally isolated [23] analogues. However, compound **3** showed roughly ten-fold stronger activity than **1**. All direct comparative studies that observed this decrease in antibacterial activity used other bacterial species such as MRSA, and *Staphylococcus epidermidis* [23], therefore, this may be a strain specific effect for the new rubrolides upon *B. subtilis*.

## 3. Materials and Methods 

### 3.1. General Procedures

UV/Vis spectra were extracted from HPLC chromatograms. A 600 MHz Varian Direct Drive spectrometer (Varian, Palo Alto, CA, USA) equipped with a 5 mm PFG dual broadband probe or a JEOL JNM-ECZ600R (JEOL, Tokyo, Japan) with a nitrogen cooled 5 mm SuperCOOL cryogenic probe were used to record the NMR spectra of all compounds (600 MHz for ^1^H nuclei and 150 MHz for ^13^C nuclei). The residual solvent peak was used as an internal reference for ^1^H (δ_H_ 2.50, DMSO–d_6_) and ^13^C (δ_C_ 39.52, DMSO–d_6_) chemical shifts [37]. Samples were quantified by ^1^H NMR spectroscopy using the residual DMSO–d_5_ peak, calibrated and acquired according to the parameters described by Pierens and co–workers [38]. Standard pulse sequences supplied by Varian and JEOL were used for NMR analyses. High–resolution (ESI) mass spectrometric data were obtained with an Agilent 6530 Accurate Mass Q–TOF LC–MS (Agilent, Santa Clara, CA, USA) equipped with a 1260 Infinity binary pump. 

Reversed–phase column chromatography was achieved using Supelco Diaion HP-20^®^ (PSDVB) (Sigma-Aldrich, Bellefonte, PA, USA) chromatographic resin. HPLC purifications were carried out using a Rainin Dynamax SD–200 solvent delivery system with 25 mL pump heads with a Varian Prostar 335 diode array detector. Octadecyl–derivatized silica (C_18_, 5 μm, 100 Å) HPLC columns (Phenomenex, Torrance, CA, USA) were either analytical (4.6 mm × 250 mm, 1 mL/min) or semi–preparative (10 mm × 250 mm, 4 mL/min). All solvents used were of HPLC grade and H_2_O was glass distilled. Solvent mixtures are reported as percent volume/volume. All reagents were of commercial quality, obtained from either Sigma Aldrich or AK Scientific and were used without prior purification. 

### 3.2. Collection of Synoicum kuranui

The tunicate *Synoicum kuranui* was hand collected using SCUBA from Great Barrier Island, NZ in 1999, as part of the NIWA collection (catalogue number NIWA 101234 and MNP0303). The tunicate was identified by MJP. Frozen tunicate was stored at −18 °C, until required.

### 3.3. Extraction and Isolation

The tunicate (24 g wet weight) was extracted in MeOH (100 mL) twice overnight. The second extract, followed by the first, were passed through a HP-20^®^ column (20 mL), pre–equilibrated in H_2_O and combined following elution. The eluent was then diluted with an equal volume of H_2_O and passed back through the column twice, followed by a 60 mL H_2_O wash. The column was then eluted with 60 mL portions of (1) 30% Me_2_CO/H_2_O, (2) 75% Me_2_CO/H_2_O, and (3) Me_2_CO (fractions A1–A3, respectively). Fractions A2 and A3 were then purified on a semi–preparative C_18_ HPLC column, using isocratic 90% MeOH/H_2_O (0.2% formic acid), generating fractions B1–B4 and C1–C7 respectively. Fractions B3 and C7 afforded solely rubrolide **1** (0.65 mg, t_R_ = 6.3 min) and **3** (0.15 mg, t_R_ = 9.9 min), respectively. Fractions B4 and C5 were further purified on an analytical C_18_ HPLC column using 80% MeOH/H_2_O (0.2% FA) to afford rubrolide *E*–**4** (0.05 mg, t_R_ = 6.3 min), *Z***–4** (0.26 mg, t_R_ = 9.9 min) and **2** (0.35 mg, t_R_ = 13.1 min). The two separate **4** samples were combined for NMR analysis. 

*Rubrolide A* (**1**) yellow film; HRESIMS *m/z* 590.7075 [M − H]^−^ (calcd. for C_17_H_7_Br_4_O_4_, 590.7083); all NMR data matches those previously reported [22].*Rubrolide B* (**2**) yellow film; HRESIMS *m/z* 624.6694 [M − H]^−^ (calcd. for C_17_H_6_Br_4_ClO_4_, 624.6694); all NMR data matches those previously reported [22].*Rubrolide T* (**3**) yellow film; UV (MeOH/H_2_O) λ_max_ 230, 250, 257 (sh), 340 nm; ^13^C and ^1^H NMR data, Table 1 and Table 2 respectively; HRESIMS *m/z* 604.7254 [M − H]^−^ (calcd. for C_18_H_9_Br_4_O_4_, 604.7240); HRESIMS/MS (50 eV) *m/z* (% relative intensity) 589.6992 (5.4), 545.7086 (3.1), 510.7802 (20.8), 482.7853 (14.5), 454.7901 (5.0), 402.8585 (8.3), 287.8473 (18.8), 272.8544 (75), 78.9188 (100).*E/Z–Rubrolide U* (**4**) yellow film; UV (MeOH/H_2_O) λ_max_ 232, 254, 361 nm; ^13^C and ^1^H NMR data, Table 1 and Table 2 respectively; HRESIMS *m/z* 526.8143 [M − H]^−^ (calcd. for C_18_H_10_Br_3_O_4_, 526.8135); HRESIMS/MS (50 eV) *m/z* (% relative intensity) 511.7898 (3.4), 483.7952 (3.0), 467.8005 (3.3), 432.8721 (12.5), 404.8767 (5.0), 375.8743 (20.0), 295.9485 (22.5), 272.8561 (50), 78.9194 (100).

### 3.4. LC– MS^2^ Analysis and Molecular Networking

The HP-20^®^ fractions (A1 and A3, 1 mg/mL in MeOH) were analyzed using the aforementioned mass spectrometer, operating in (−)-polarity at a mass range of *m/z* 50–1500. Instrumental parameters for data acquisition were set as follows: capillary voltage of 3500 V, nebulizer gas (N_2_) pressure of 30 psig, ion source temperature of 275 °C, sheath gas temperature of 300 °C and flow of 7 L/min, and the acquisition rate was three spectra/s. The ion isolation width was set wide at 7 amu. Minutes 0–0.5 were sent to waste and minutes 0.5–25 recorded with untargeted ion fragmentation (auto–MS^2^), where the collision induced dissociation (CID) energies were dependent on the precursor mass determined by the pre–set equations CID = 2.62*x* + 14.75 (low energy) and CID = 3.93*x* + 22.13 (high energy). The five most intense ions per MS scan were subjected to CID and were actively excluded after three spectra for 0.3 min.

The samples were injected (injection volume: 10 µL) into the system equipped with an Eclipse Plus reversed–phase C_18_ column (30 mm × 2.1 mm, 3.5 *μ*m; Agilent Technologies) at 35 °C. Compound separation was achieved using mobile phase A 99.9% H_2_O/0.1% NH_4_HCO_2_ and B 99.9% ACN/0.1% FA, at 0.4 mL/min with the following gradient method: 0–1 min 2.5% B, 1–20 min 2.5%–100% B, 20–25 min 100% B and a column re–equilibration period for 3 min. 

The LC–MS^2^ data was converted to MGF files using Agilent’s Masshunter (qualitative analysis software, B.04.00) and uploaded to GNPS, which was used to create a molecular network [1]. MS^2^ fragment ions within ± 17 Da of the parent *m/z* were removed. MS^2^ spectra were then window filtered, selecting the six largest peaks within a window of 50 *m/z* before clustering with MS-Cluster using tolerances of 2.0 Da for the parent and 0.5 Da for MS^2^ ions, to generate consensus spectra, although any consensus spectrum containing only 0–2 spectra was removed. Filtering of network edges to giving > four matched peaks, with cosine score >0.6, was applied to form the ultimate network. Only those edges connecting two nodes that appeared within its partner top 10 most similar nodes were retained within the network. Online spectral databases stored within GNPS were used for node dereplication/annotation; all library fragmentation data were filtered identically as experimental data. Matches between library and the experimental data were kept only if a cosine score greater than 0.7 was obtained, with a minimum of six matching ions. The experimental data can be obtained from: https://gnps.ucsd.edu/ProteoSAFe/status.jsp?task=0faf2a77b21b457c9a047343e9b813e0#. All MNs were viewed using Cytoscape 3.7.0. [39].

### 3.5. Antibacterial Bioassay

Minimum inhibitory concentration (MIC) studies were performed using an established protocol [40] to test isolated compounds against test strains of *E. coli* BL21 and *B. subtilis* 168. All tests were carried out in MH medium at 30 °C, with 16 h growth time in biological triplicates (n = 3 independent experiments). A positive control of tetracycline and negative control DMSO were included in the analysis. All compounds were assayed from 128 µg/mL in two-fold serial dilution steps across 10 dilution stages to the lowest concentration tested 0.25 µg/mL. Sterility (media only) and growth (cells only) controls were included in each plate assayed.

## 4. Conclusions

GNPS is a powerful tool for the detection and discovery of new variations of previously reported metabolites within a complex chemical extract. Although the rubrolides have been thoroughly investigated, the MN revealed two previously unreported precursor ions, which upon a targeted isolation, led to the isolation of new rubrolides T (**3**) and *E*/*Z*-U (**4**) Based on the difference in *m/z* of the neighboring precursor ions, the molecular formula was predicted, which streamlined the ensuing NMR spectroscopy–based structure elucidation. Both **3** and **4** showed strong growth inhibition against *B. subtilis*, but no activity against *E. coli*. This finding is significant in that it differs from a lack of biological activity for methylated rubrolides against Gram positive bacteria previously reported and is worthy of future investigation. This work reflects how MN can be used to probe complex extracts for the presence of new bioactive, albeit minor, compounds.

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
