# Peer review of "Targeted Isolation of Rubrolides from the New Zealand Marine Tunicate *Synoicum kuranui"

_marinedrugs, 2020, doi:10.3390/md18070337_

Round 1
Reviewer 1 Report
This present manuscript describes the isolation of new rubrolides, a difficult task given the extensive prior investigation into this structural class. Overall, this study is a strong example of how modern tools can be brought to bear on natural product isolation. The introduction was well-referenced and described well the key methodology as well as the importance of these natural products. The characterization was clearly described and well-supported. The biosynthetic pathway and E/Z isomerization of compounds 4 were both plausible, and in the former case drawn from precedent of related natural products.
Author Response
We are grateful to the referee for their very kind words. We believe no changes were required.
Reviewer 2 Report
Well structured and concise manuscript.
I recommend to extend the conclusions section by briefly stating the insights towards biological activity regarding the new compounds.
To focus some of the statements and correct some minor errors I suggest the following edits:
Line 19-20 "observed" can be cropped and I believe "unreported" should mean "reported"
32-33 maybe rephrase, e.g. change order of "in an extract" and "within a structure"
51-55 mind recurrent use of "which", maybe split statements into two sentences
59 I think one could likewise assume, that given many known compounds there may be many unknowns too. Maybe rather reiterate here that MN can assist to find new NPs even in well studied organisms.
64 remove "a"
67 Presented MN was visualized in Cytoscape and created using GNPS. Best insert a reference for Cytoscape (I believe Shannon et al. 2003)
Fig.1 indicate which parameter determines relative contribution to nodes (e.g. sum precursor intensity). Also specify what exactly the values in the figure refer to (e.g. I assume monoisotopic precursor ion m/z values are annotated to nodes. For edges, most authors give cosine values between 0 and 1. Label accordingly or specify the given unit.
131 isomeric ions are not identical, but ion masses are
172 and 176 include reference or indicate if hypothesized
Scheme 1 mind direction of the bottom reaction arrow, maybe for completeness include H+ in the bottom left structure
197 unbold characters after "4"
254 remove "by"
274 check "psig"
284 maybe simply state 1-25 min 100% B
286 insert a word after "files"
300 cite Cytoscape paper here if not in line 67
Supplementary Check Table of Contents vs Figure legends (e.g. Fig. S3 and Table S2). Mind spelling of species names in Fig. S1, it would also be nice to include the GNPS link here if possible.
52 e.g. change "currently unreported" for "new compounds"
Interesting work, well done!
Author Response
Well structured and concise manuscript.
RESPONSE: We thank the reviewer for their kind words. We are particularly impressed with the vigilance the referee employed in finding many small but significant omissions in our initial submission.
I recommend to extend the conclusions section by briefly stating the insights towards biological activity regarding the new compounds.
RESPONSE: We have added the following text “Both 3 and 4 showed strong growth inhibition against B. subtilis but no activity against E. coli. This finding is significant in that it differs from a lack of biological activity for methylated rubrolides against Gram positive bacteria previously reported and is worthy of future investigation.”
To focus some of the statements and correct some minor errors I suggest the following edits:
Line 19-20 "observed" can be cropped and I believe "unreported" should mean "reported"
RESPONSE: Done. We appreciate the referee picking up this error in our text.
32-33 maybe rephrase, e.g. change order of "in an extract" and "within a structure"
RESPONSE: Done.
51-55 mind recurrent use of "which", maybe split statements into two sentences
RESPONSE: Done.
59 I think one could likewise assume, that given many known compounds there may be many unknowns too. Maybe rather reiterate here that MN can assist to find new NPs even in well studied organisms.
RESPONSE: Done.
64 remove "a"
RESPONSE: Done.
67 Presented MN was visualized in Cytoscape and created using GNPS. Best insert a reference for Cytoscape (I believe Shannon et al. 2003)
RESPONSE: We have referenced Cytoscape in the experimental section as suggested below, but not here.
Fig.1 indicate which parameter determines relative contribution to nodes (e.g. sum precursor intensity). Also specify what exactly the values in the figure refer to (e.g. I assume monoisotopic precursor ion m/z values are annotated to nodes. For edges, most authors give cosine values between 0 and 1. Label accordingly or specify the given unit.
RESPONSE: Done.
131 isomeric ions are not identical, but ion masses are
RESPONSE: Done.
172 and 176 include reference or indicate if hypothesized
RESPONSE: Done.
Scheme 1 mind direction of the bottom reaction arrow, maybe for completeness include H+ in the bottom left structure
RESPONSE: Done.
197 unbold characters after "4"
RESPONSE: Done.
254 remove "by"
RESPONSE: Done.
274 check "psig"
RESPONSE: PSIG is the correct term, referring to “pounds per square inch gauge” pressure, check with the manufacturer.
284 maybe simply state 1-25 min 100% B
RESPONSE: Added “2.5%” before “100%” to indicate a gradient separation profile.
286 insert a word after "files"
RESPONSE: Done.
300 cite Cytoscape paper here if not in line 67
RESPONSE: Done.
Supplementary Check Table of Contents vs Figure legends (e.g. Fig. S3 and Table S2). Mind spelling of species names in Fig. S1, it would also be nice to include the GNPS link here if possible.
RESPONSE: Without specific mention of what the referee had noticed, we are unsure what they would like us to change. We have purposefully shorted each entry by removing NMR field strength and solvent used to maintain a concise list. All page numbers match correctly to the best of our knowledge.
52 e.g. change "currently unreported" for "new compounds"
RESPONSE: Done.
Interesting work, well done!
Reviewer 3 Report
The article "Targeted Isolation of Rubrolides from the New
Zealand Marine Tunicate Synoicum kuranui" by Bracegirdle et al. describes GNPS guided identification and isolation of two new rubrolides from S. kuranui. The autors presented plausible explanation for formation and isomerization of rubrolides. The isolated compounds were screened for their antimicrobial activity; however, only E. coli and B. subtilis were used for the study. Interestingly, E. coli was not influenced by rubrolides, whereas the growth of B. subtilis was inhibited with rubrolides at submicromolar concentrations. The article can be even more attractive, if the authors shows inhibition of S. aureus or MRSA with these rubrolides. Nevertheless, the article deserves to be published as it is.
Author Response
We are grateful to the referee for their very kind words. We have been unable to test our compounds against MRSA at this time due to a paucity of material, but as this is an ongoing study we will attempt to include that strain in subsequent bioassays.